# Epilepsy detection based on multi-head self-attention mechanism

**Yandong Ru**[1,2]*, **Gaoyang An**[3], **Zheng Wei**[3], **Hongming Chen**[1,2]

**1** Key Laboratory of Oceanographic Big Data Mining & Application of Zhejiang Province, Zhoushan, China,
**2** School of Information Engineering, Zhejiang Ocean University, Zhoushan, China, **3** Heilongjiang University of Science and Technology, Harbin, China

* 2023005@zjou.edu.cn

## Abstract

CNN has demonstrated remarkable performance in EEG signal detection, yet it still faces limitations in terms of global perception. Additionally, due to individual differences in EEG signals, the generalization ability of epilepsy detection models is week. To address this issue, this paper presents a cross-patient epilepsy detection method utilizing a multi-head self-attention mechanism. This method first utilizes Short-Time Fourier Transform (STFT) to transform the original EEG signals into time-frequency features, then models local information using Convolutional Neural Network (CNN), subsequently captures global dependency relationships between features using the multi-head self-attention mechanism of Transformer, and finally performs epilepsy detection using these features. Meanwhile, this model employs a light multi-head attention mechanism module with an alternating structure, which can comprehensively extract multi-scale features while significantly reducing computational costs. Experimental results on the CHB-MIT dataset show that the proposed model achieves accuracy, sensitivity, specificity, F1 score, and AUC of 92.89%, 96.17%, 92.99%, 94.41%, and 96.77%, respectively. Compared to the existing methods, the method proposed in this paper obtains better performance along with better generalization.

## Introduction

Epilepsy, as a common neurological disorder, is characterized by chronicity, abruptness, and recurrence [1]. According to statistics from the World Health Organization, there are over 50 million epilepsy patients worldwide, with nearly 80% in low and middle-income countries [2], significantly impacting patients' work and life. Due to the fact that electroencephalogram (EEG) signals contain a wealth of physiological and pathological information from patients, they are widely used for detecting epileptic seizures [3]. However, in-depth analysis of these EEG signals to accurately determine whether an epileptic seizure is occurring is a laborious task that relies on experience, which can lead to misdiagnosis or missed diagnosis. Therefore, constructing epilepsy detection models using artificial intelligence technology holds significant theoretical and practical value.

The current mainstream epilepsy detection models can be roughly divided into two categories: machine learning models and deep learning models. Machine learning models typically

**Data Availability Statement:** The data underlying the results presented in the study are available from (https://physionet.org/content/chbmit/1.0.0/).

**Funding:** this research was supported by the Talent Fund of Zhejiang Ocean University (Project No: JX6311061523). The funders had no role in study

design, data collection and analysis, decision to publish, or preparation of the manuscript.

**Competing interests:** The authors have declared that no competing interests exist.

rely on manually extracted features for epilepsy detection. This approach is limited by dataset variations, making it challenging to ensure accuracy. In contrast, deep learning models can directly learn hierarchical and abstract features from EEG signals, which are more discriminative than traditionally handcrafted features, simplifying the feature extraction process and improving detection accuracy.

In the field of epilepsy detection using deep learning, Convolutional Neural Networks (CNN) are the most widely applied. For instance, the epilepsy seizure automatic detection model proposed by A. Gramacki et al. [4], adopts an end-to-end learning approach using CNNs. In such models, CNNs directly learn features from raw EEG signals and output detection results without the need for manual feature extraction or selection. S. Khalilpour et al. [5] based their work on 1D-CNN and combined it with a channel selection strategy for epilepsy signal detection. This method primarily achieves detection by optimizing the network structure and the quality of input data. M.S. Hossain et al. [6] designed a 2D-CNN model to capture EEG signal spectral and temporal features, utilizing these captured features for epilepsy signal detection. However, in epilepsy detection tasks, the issue of data imbalance due to the scarcity of seizure events can affect the performance of epilepsy detection. Therefore, I. Ahmad et al. [7] used K-means oversampling techniques to balance the data, integrating 1D CNNs with Bi-LSTM networks to efficiently extract spatiotemporal information. H. Fei et al. [8] proposed an epilepsy detection method that combines imbalanced classification with deep learning. They balanced EEG data through data augmentation and designed a 1D CNN for efficient detection. Moreover, CNN variants [9, 10] and hybrid models [11–13] are also gradually being applied in the field of epilepsy signal detection.

However, CNNs are limited by the size of the convolutional kernels, leading to restrictions in capturing global dependencies. They overlook the continuity of adjacent time segments, fail to comprehensively consider the global changes in signals, and their learning process is always unidirectional. Transformers' self-attention mechanism [14] addresses this limitation by capturing long-distance dependencies in signals and enabling parallel computation. Therefore, researchers have begun to explore applying attention mechanisms to EEG signals. For instance, Z. Li et al. [15] improved a Transformer-based recognition model used for identifying gesture EEG signals. Y. Song et al. [16] proposed a method for decoding motor imagery EEG based on FBCSP (Filter Bank Common Spatial Patterns) and Transformer models.

However, due to the limitations of Transformer models in extracting local information using the self-attention mechanism, CNNs with excellent local feature extraction capabilities are combined to compensate for this deficiency. In a study by J. Sun et al. [17], they constructed multiple hybrid network models based on Transformers for motor imagery (MI) EEG classification, demonstrating that combining attention mechanisms with CNNs yielded better results. H. Liu et al. [18] proposed a 1D-CNN based on Transformers (CNN-Transformer) for identifying motor imagery EEG signals, showing superior performance compared to the classical CNN-LSTM architecture. Y. Song et al. [19] introduced a method named EEG Conformer, achieving advanced performance in motor imagery and emotion recognition paradigms based on EEGs. In the field of epilepsy detection, X. Deng et al. [20] proposed an epilepsy detection model based on CNNs and attention mechanisms. This model extracts EEG features using CNNs and reinforces these features using attention mechanisms for efficient classification. Y. Sun et al. [21] proposed an end-to-end model that includes both convolutional layers and Transformer layers. Through experiments, they demonstrated that combining CNN with Transformer can enhance the performance of epilepsy detection models.

Additionally, there are significant individual differences in EEG signals, although one-to-one epilepsy detection methods are more accurate, they cannot be shared across patients. Training a model for each patient is impractical. Therefore, researching cross-patient epilepsy

detection models is more practical. For instance, S. Yang et al. [22] trained a cross-patient epilepsy detection model using EEG data from multiple patients. It eliminates the need to model each patient individually, allowing a single model to detect epilepsy in different patients. Z. Wang [23] proposed a method called SDS-GDA-TASA, experimental results show that this method outperforms various existing methods in cross-patient seizure classification. However, there is currently no research on using a hybrid network model for cross-patient epilepsy detection constructed based on the self-attention mechanism.

Incorporating the aforementioned analysis, this paper investigates a cross-patient epilepsy detection method. The main contributions of this paper are as follows: (1) This paper proposes an epilepsy detection method based on a multi-head attention mechanism for the purpose of detecting epileptic seizures. The proposed method employs a CNN module to extract local feature relationships from EEG sequence data, while leveraging a multi-head self-attention mechanism to capture global dependency relationships, thereby overcoming the limitations of CNNs and maximizing the benefits of the multi-head self-attention mechanism. (2) An improved Transformer module is proposed, consisting of a CNN layer, a light multi-head self-attention layer, and a residual feed-forward network. This effectively improves computational efficiency and reduces training time. (3) Cross-individual experiments conducted on the CHB-MIT dataset demonstrate that the model proposed in this paper exhibits excellent performance in epileptic seizure detection, effectively overcoming variations in seizure patterns among different patients and enabling cross-patient epileptic seizure detection.

The organization of the subsequent sections of this paper is as follows: The Methods section details the proposed method; The Experiments section provides a detailed description of the experimental setup and result analysis; The Conclusion section concludes this paper.

## Methods

The CNN is a deep feed-forward neural network characterized by local connectivity and weight sharing. It stands as one of the representative algorithms in the realm of deep learning. CNN excels in local feature extraction and finds widespread use in the detection of epileptic signals. The multi-head self-attention mechanism, a crucial component of the Transformer model, calculates the degree of correlation between different locations in the input sequence, producing corresponding weighted vector representations. The multi-head self-attention mechanism of the Transformer captures dynamic correlation and long-distance dependence between EEG signals, addressing the challenge of the weak global feature extraction ability of CNN. This paper introduces a new hybrid network model by fusing CNN and Transformer. The model maximizes the utilization of correlation between local and global features. Initially, the raw EEG signals undergo processing using signal processing techniques. Subsequently, CNN is employed to extract local features. Then the multi-head self-attention mechanism is utilized to learn global features of the signals. Finally, these features are employed for epilepsy detection. The structure is illustrated in Fig 1.

As shown in Fig 1, The model comprises four main stages. In the first stage, the original EEG signal undergoes pre-processing using signal processing techniques to filter out mixed noise and ensure uniformity in sample data dimensions. In the second stage, the focus is on extracting short-term temporal patterns of the EEG signal and local dependencies between individual EEG channels. The third stage uses the self-attention mechanism to capture long-range dependencies and temporal dynamic correlations between different feature vectors. The fourth stage concludes with the global average pooling layer and the fully connected (FC) layer. The final results of epilepsy detection are obtained using the logsoftmax function.

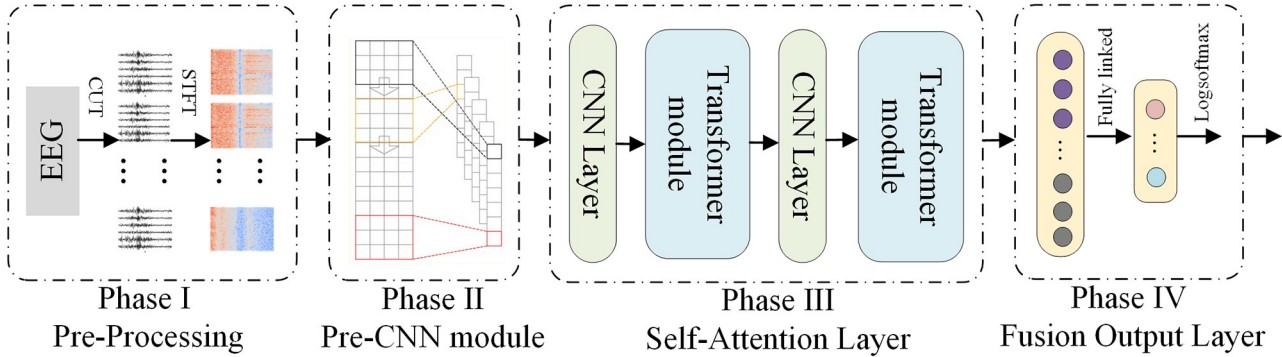

**Fig 1. Model structure.**

### Front convolutional layer

The Convolution layer, at the heart of the convolutional neural network model, serves as a feature extractor. Through the convolution operation, this layer automatically extracts local signal features, eliminating the need for manual feature extraction and potential mismatches, thereby enhancing model performance. The process of the convolution operation is illustrated in Fig 2.

The convolution kernel slides over each part of the input in a given step size and operates through the convolution in order to extract different features and finally produce a feature map. In a convolutional layer, the convolution kernel transforms the output of the previous layer and inputs the result of the transformation into a nonlinear activation function to construct the output features. The general form of convolution operation is as shown in Eq (1).

$$x_j^l = y(\sum_{i \in M_j} x_i^{l-1} * w_{ij}^l + b_j^l)$$ (1)

In Eq (1), $x_i^{l-1}$ is represents the convolution region corresponding to the $i$ convolution kernel in the $l-1$ layer, $x_i^l$ is the $j$ feature map in the $l$ layer, $M$ is the set of input features, $w$ is the weight matrix of the convolution kernel, $b$ is the bias, $y$ is the activation function, and $*$ is the convolution operation.

This paper employs the pre-CNN module for extracting local information, as illustrated in Fig 3.

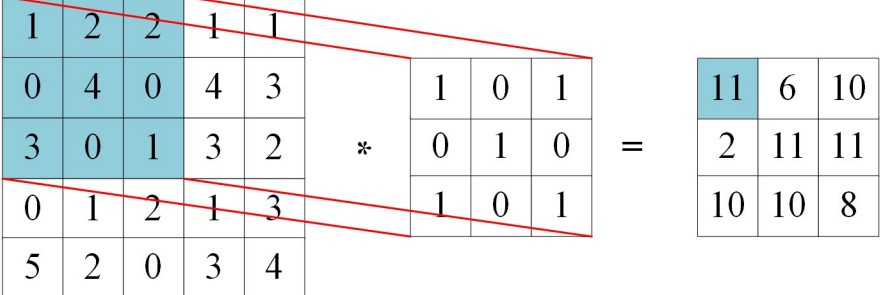

**Fig 2. Convolution operation with step size 1.**

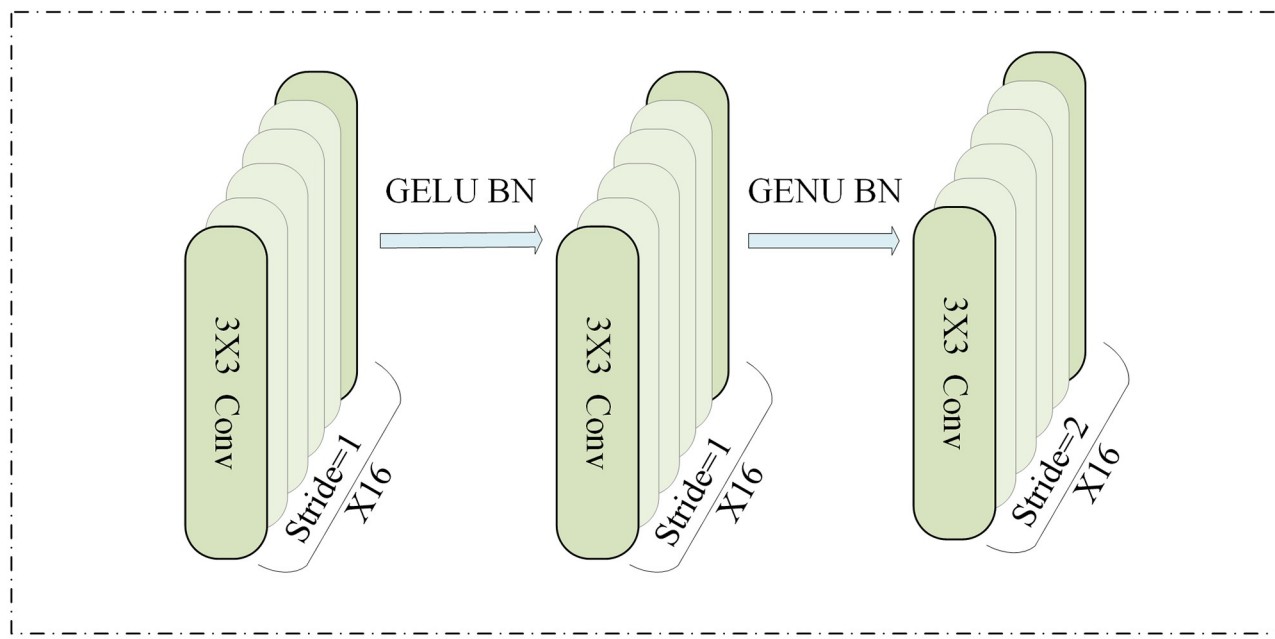

**Fig 3. Pre-CNN module.**

This module comprises three convolutional layers, each with a 3×3 convolutional kernel. The total number of convolutional kernels is 16. The first two convolutional layers use a step size of 1, while the last convolutional layer has a step size of 2. By stacking 3x3 convolutional kernels to gradually increase the receptive field, the expressive power of the model is enhanced. The first two convolutional layers use a stride of 1, preserving the spatial information's fine details, which helps capture finer features. The last convolutional layer uses a stride of 2 to achieve feature map dimension reduction, enabling the model to focus more on global and abstract features. This design aims to extract short-term temporal patterns of EEG signals and local dependencies between individual EEG channels. Simultaneously, it captures temporal sequence features with translation-invariant compression to address the gradient vanishing problem. Following each convolutional layer is a GELU activation and batch normalization layer. This arrangement aims to diminish internal covariate bias during neural network training, mitigate the risk of overfitting, and enhance convergence speed. Subsequent to the nonlinear activation function, a dropout operation is incorporated to minimize complex coadaptation between neurons.

## Multi-head self-attention module

The feature representation acquired through convolution exhibits translational invariance in the time dimension. Moreover, it manifests long-range dependencies and temporal dynamic correlations among different feature vectors. Incorporating the multi-head self-attention mechanism facilitates the learning of diverse information in distinct subspaces. This, in turn, enhances global feature extraction capability and temporal dynamic correlation. Additionally, the multi-head self-attention mechanism assigns distinct weights to various feature vectors, effectively resolving the issue of equal contribution from each feature vector. The multi-head self-attention module comprises two phases, each featuring a combination of a CNN layer and

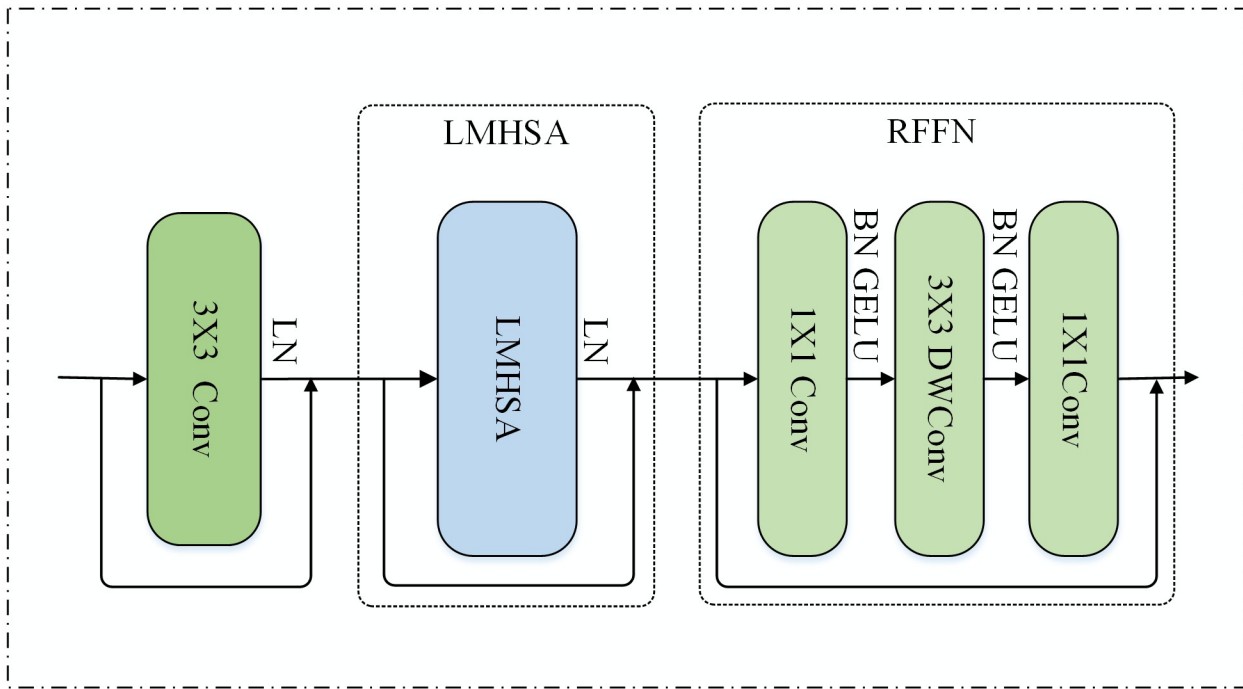

**Fig 4. Multi-head self-attention module.**

a Transformer block designed for extracting long-term dependencies. One of the stages is depicted in Fig 4.

Every CNN layer comprises a 3×3 convolutional layer and a layer normalization (LN) layer. This configuration aims to reduce the feature dimensions of intermediate features while preserving the translational invariance of the information. Subsequently, there is a light multi-head self-attention layer (LMHSA) for global feature extraction and a residual feed forward network (RFFN) for improved nonlinear representation. Two stages of Transformer blocks are arranged in a stacked, alternating structure. This arrangement aids in capturing more comprehensive multi-scale features.

**Translation invariance.** Translation invariance stands as a crucial property in classification tasks [24]. Nevertheless, the absolute positional coding employed in the original Transformer compromises this property [25]. Therefore, in this paper, CNN is employed to maintain translation invariance. The convolution equations is shown in Eq (2).

$$X' = Conv(X) + X \tag{2}$$

In Eq (2), $X \in R^{H \times W \times D}$, $H$ and $W$ respectively represent the height and width of the input feature matrix $X$, $D$ represents the dimension of the feature, $Conv()$ represents the convolution, and $X'$ represents the output.

**Light multi-head self-attention mechanism.** The self-attention mechanism primarily employs the product operation of scaling points. The attention module takes input from three matrices: query (Q), key (K), and value (V). The output is a weighted sum of similarity and value based on Q and K. The original self-attention mechanism poses computational intensity challenges due to the high dimensionality of K and V. To alleviate computational and memory demands, this paper diminishes the spatial dimensionality of K and V by employing a 3×3

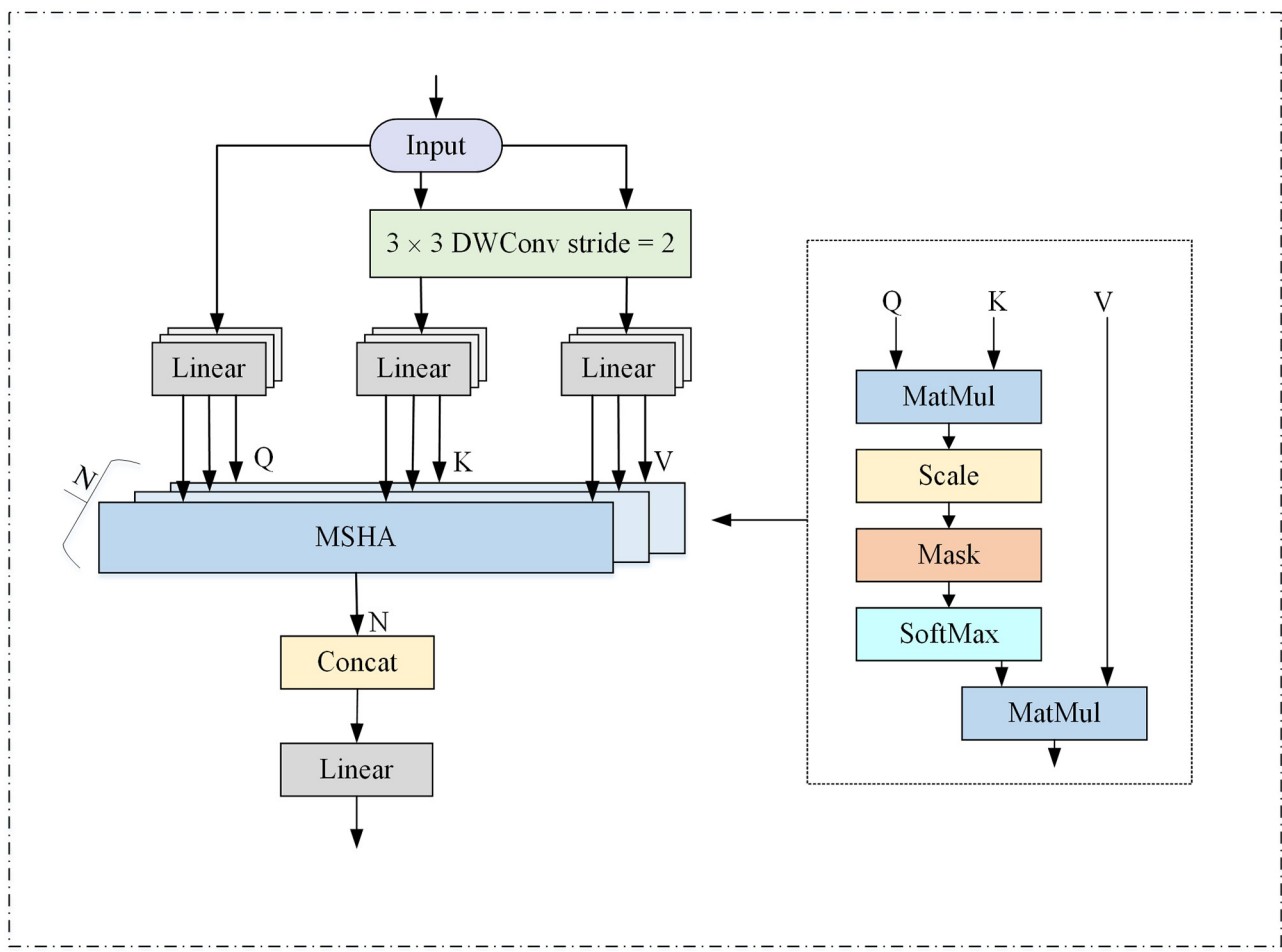

**Fig 5. Light multi-head self-attention.**

depth separable convolution with a step size of 2. In epilepsy detection tasks, this strategy reduces dimensions while minimizing information loss, thereby maintaining good performance of the model. The specifics are illustrated in Fig 5.

The equations for calculating Q, K, and V in the Fig 5 are shown in Eqs (3), (4) and (5).

$$Q = Linear(X')   \tag{3}$$

$$K = Linear(Conv(X'))   \tag{4}$$

$$V = Linear(Conv(X'))   \tag{5}$$

To enable the model to learn information from various representation subspaces, the self-attention operation is iterated multiple times by multiple parallel heads. Each head focuses on different information, allowing diverse parts of the feature representation to be processed. This facilitates the extraction of richer features and global dependencies. The self-attention results from each head are concatenated, and multi-space fusion is executed using matrix $W$. Eventually, the ultimate outcome of multi-head self-attention is derived. Specific operational

expressions are shown in Eqs (6) and (7).

$$head = Attention(Q, K, V) = soft \max(QK^T/\sqrt{d_k})V \tag{6}$$

$$X'' = sMHSA = Concact[head_1; head_2 \ldots \ldots head_n]W^O \tag{7}$$

In Eqs (6) and (7), LMHSA represents the operation of the light multi-head self-attention mechanism, *soft* max represents the *soft* max function, $d_k$ represents the dimension of K, and $X''$ represents the output of the lightweight multi-head self-attention layer.

**Residual feedforward neural networks.** In the original Transformer, the feedforward neural network comprises a fully connected layer with two linear transformations and a ReLU activation function. This paper employs a residual feedforward neural network to substitute the original feedforward neural network, aiming to enhance computational efficiency and further improve the nonlinear representation of the model. The residual feedforward neural network incorporates two 1×1 convolutions and a 3×3 depth-separable convolution. Additionally, residual connections in residual networks, by directly adding the input to the output, help alleviate the vanishing gradient problem, enabling the network to learn feature representations more deeply during training. The specific structure is shown in Fig 4. The residual feed-forward neural network layer takes the output of the lightweight polytope self-attention layer $X''$ as input. As shown in Eq (8).

$$X''' = RFFN(X'') = Conv(DWConv(Conv(X''))) + X'' \tag{8}$$

In Eq (8), *RFFN* represents the operation of the residual feed forward neural network layer, *DWConv*() represents depth-wise separable convolution, and $X'''$ is the output of the residual feed forward neural network layer.

## Fusion output layer

The global pooling layer executes the pooling operation on the feature representation to derive the coded sequence post the fully connected layer. The internal logsoftmax function calculates the probability value for the corresponding category. The probability value is binarized for classification based on the set threshold, yielding the final classification result. Calculated as shown in Eq (9).

$$p = Logsoft \max\left(W_P h_{t+1} + b_p\right) \tag{9}$$

In Eq (9), $W_p$ and $b_p$ respectively represent the weight matrix and bias term.

## Experiments

### Performance evaluation indicators

To ensure the rigor of the experimental results, this paper examines the performance of the model by selecting sensitivity (SEN), accuracy (ACC), specificity (SPE), area under the curve (AUC), and F1 score (F1-Score).

Accuracy is the ratio of the number of correctly classified samples by the classifier to the total number of samples. Calculated as shown in Eq (10).

$$ACC = \frac{TP + TN}{TP + TN + FP + TN} \tag{10}$$

Sensitivity is the proportion of samples that are actually positive and are correctly identified

as positive. Calculated as shown in Eq (11).

$$SENS = \frac{TP}{TP + FN} \tag{11}$$

Specificity evaluates the performance of the classifier in identifying negative cases. Calculated as shown in Eq (12).

$$SPE = \frac{TN}{TN + FP} \tag{12}$$

F1 score is the harmonic mean of precision and recall. Calculated as shown in Eq (13).

$$F1 - Score = 2 \times \frac{precision \times recall}{precision + recall} \tag{13}$$

In Eq (13), recall is the same as sensitivity, the formula for precision is as follows:

$$precision = \frac{TP}{TP + FP} \tag{14}$$

AUC is the area enclosed by the ROC curve and the coordinate axes, where the horizontal axis of the ROC curve represents the false positive rate (FPR), and the vertical axis represents the true positive rate (TPR). The calculation for AUC, FPR and TPR are as follows:

$$AUC = \frac{1}{2} \sum_{i=1}^{n-1} (x_{i+1} - x_i)(y_i + y_{i+1}) \tag{15}$$

$$FRP = \frac{FP}{FP + TN} = x \tag{16}$$

$$TPR = \frac{TP}{TP + FN} = y \tag{17}$$

In Eq (15), $(x, y)$ represent the continuous coordinates on the ROC curve.

In Eqs (10), (11), (12), (14), (16) and (17), TP represents true positives, FN represents false negatives, FP represents false positives, and TN represents true negatives.

## Data set

The dataset utilized in this paper is sourced from the publicly available epileptic EEG dataset recorded and curated by researchers affiliated with Children's Hospital Boston (CHB) and the Massachusetts Institute of Technology (MIT), known as CHB-MIT [26]. A total of 22 epilepsy patients (5 males, 3–22 years old; 17 females, 1.5–19 years old) were recorded in CHB-MIT. The epileptic EEG signals in the CHB-MIT dataset were sampled at a frequency of 256 HZ, with a 16-bit sampling resolution. In the majority of cases, 23 leads were utilized. Given that many patients in the CHB-MIT dataset experienced multiple consecutive epileptic seizures, this paper categorized intervals of consecutive seizures lasting less than 30 minutes as a single seizure. The interictal period was defined as a time interval of at least 4 hours before and after the seizure state. In addition, this paper only selected participants who had no more than 10 seizures in 24 hours. Combining the above definitions and the actual situation, the total number of epileptic patients who satisfied the above conditions was 16, as depicted in Table 1.

**Table 1. CHB-MIT data set.**

| No. of patients | Gender | Age | Electrode | No. of seizures | Seizures total duration(s) | Total duration(h) |
|---|---|---|---|---|---|---|
| Chb1 | Female | 11 | 23 | 7 | 442 | 40.55 |
| Chb2 | Male | 11 | 23 | 3 | 172 | 35.26 |
| Chb3 | Female | 14 | 23 | 7 | 402 | 38.00 |
| Chb5 | Female | 7 | 23 | 5 | 558 | 39.00 |
| Chb8 | Male | 3.5 | 23 | 5 | 919 | 20.00 |
| Chb9 | Female | 10 | 23 | 4 | 276 | 67.87 |
| Chb10 | Male | 3 | 23 | 7 | 447 | 50.00 |
| Chb13 | Female | 3 | 23 | 10 | 440 | 11.00 |
| Chb14 | Female | 9 | 23 | 8 | 169 | 26.00 |
| Chb16 | Female | 7 | 23 | 8 | 69 | 17.00 |
| Chb17 | Female | 12 | 23 | 3 | 293 | 20.00 |
| Chb18 | Female | 18 | 23 | 6 | 317 | 34.64 |
| Chb19 | Female | 19 | 23 | 3 | 236 | 28.92 |
| Chb20 | Female | 6 | 23 | 8 | 296 | 27.6 |
| Chb21 | Female | 13 | 23 | 4 | 199 | 32.82 |
| Chb23 | Female | 6 | 23 | 7 | 424 | 26.51 |

## Data preprocessing methods

In epilepsy detection tasks, defining epileptic EEG data into two states (seizure and non-seizure)—is an important preprocessing step that helps deep learning models accurately classify epileptic states. Fig 6 shows the EEG signal waveforms for these two states.

From Fig 6, it can be observed that there is an imbalance in the amount of seizure data. Moreover, given the variability in the length of each record within the case files and the likelihood of multiple seizures within seizure records, further data preprocessing was undertaken to ensure uniformity in sample data dimensions. The original EEG data was initially partitioned into consecutive 30-second windows. From these windows, time-frequency features were extracted to analyze the EEG signals. Additionally, the independent component analysis method was applied to filter out noise present in the EEG signal. This paper employed the STFT to convert the processed time series EEG signals into spectrograms, which preserve the important information of the original EEG signals. The spectrogram calculated from a segment of EEG signals in the CHB-MIT dataset is shown in Fig 7.

The horizontal axis of this spectrogram represents frequency, displaying the different frequency components contained in the signal. The vertical axis represents time, displaying how the signal changes over time. The color of the spectrogram represents the intensity of the signal at specific time and frequency points, with darker colors indicating higher signal intensity at those time and frequency points. Fig 7 shows a significant energy distribution within approximately the 0~40Hz frequency range over a 30-second window, which aligns with the conventional focus on low-frequency bands in EEG analysis. The STFT equations is shown in Eq (18).

$$STFT(x(t))(\omega, \tau) = \int_{-\infty}^{\infty} x(t)\omega(t - \tau)e^{-i\omega t}dt \tag{18}$$

In Eq (18), $x(t)$ represents the signal to be transformed, and $\omega(t)$ is the Gaussian window function.

To address the data imbalance problem in the epilepsy detection task, this paper adopts the oversampling technique to process the data. By increasing the number of minority class

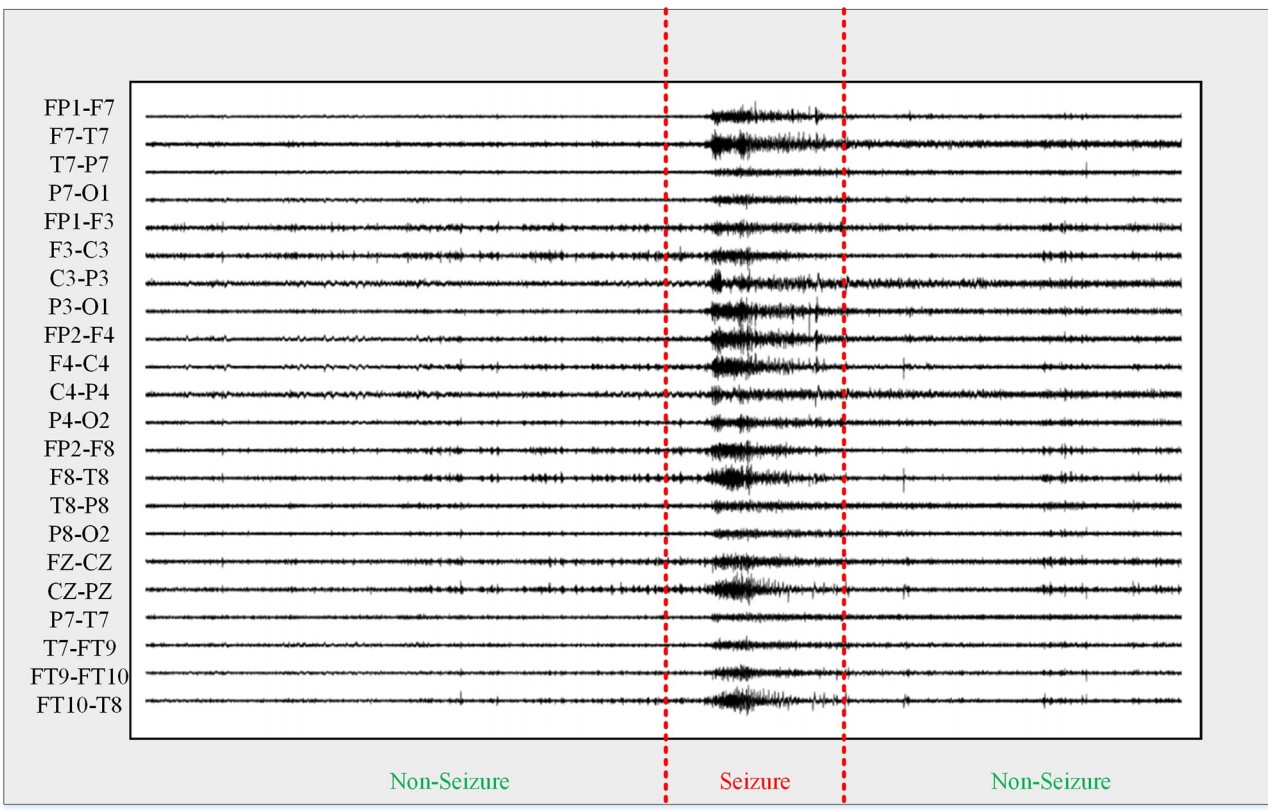

**Fig 6. Epileptic EEG signals.**

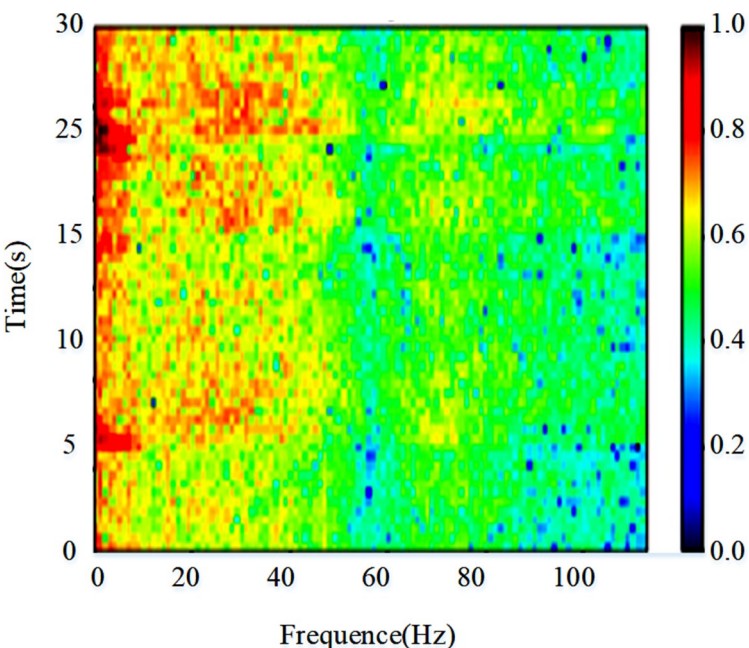

**Fig 7. Epileptic EEG signal spectrogram.**

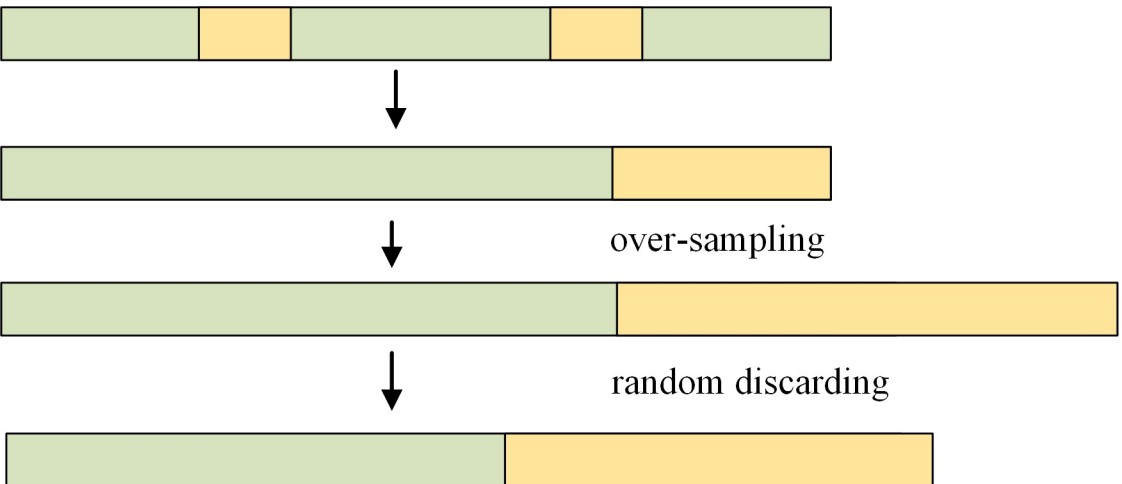

**Fig 8. Graphical representation of data processing.**

samples, oversampling techniques can assist classification models in better learning the features of minority classes, thereby improving classification performance. This method is applied individually to each patient in the dataset, ensuring an equal number of samples in both categories for model training. Given variations in patient profiles and available samples, 80% of the dataset constitutes the training set, while the remaining 20% is allocated to the test set. The specific flow is shown in Fig 8.

As shown in Fig 8, oversampling is applied to imbalanced data to generate more samples for the minority class, followed by random dropout to maintain their equivalence. This strategy introduces some randomness, which helps the model generalize better during training.

## Experimental parametric

This experiment was conducted on a platform equipped with an RTX 4060Ti GPU and Cuda 10.2 environment. Python 3.9 was used as the programming language, and the PyTorch 2.0.0 deep learning framework was employed for model construction, parameter tuning, and experimental performance evaluation. In this paper, the batch size is set to 32, the learning rate is set to 0.01, and the number of self-attention heads is set to 8. We used the binary cross-entropy loss function and chose the Adam optimizer.

**Loss function analysis.** The loss function is a function used to evaluate the degree of error between the model's detection results and the actual results, and is crucial for the model's detection performance. Comparing the performance of different loss functions in the same model helps in selecting the most suitable loss function for the current problem, thereby improving the model's performance. This paper selected three common classification loss functions (binary cross-entropy loss, focal loss, and hinge loss) for experimental comparison. The model accuracy under different loss functions is shown in Fig 9.

From Fig 9, it can be seen that compared to other loss functions, the model performs best when using binary cross-entropy loss as the loss function. This is because in classification problems, cross-entropy loss function can more directly measure classification performance and has been optimized for the characteristics of classification problems. Therefore, this paper selects binary cross-entropy loss as the loss function. Additionally, from Fig 9, it can be observed that when the training epochs reach 50, the model accuracy using binary cross-

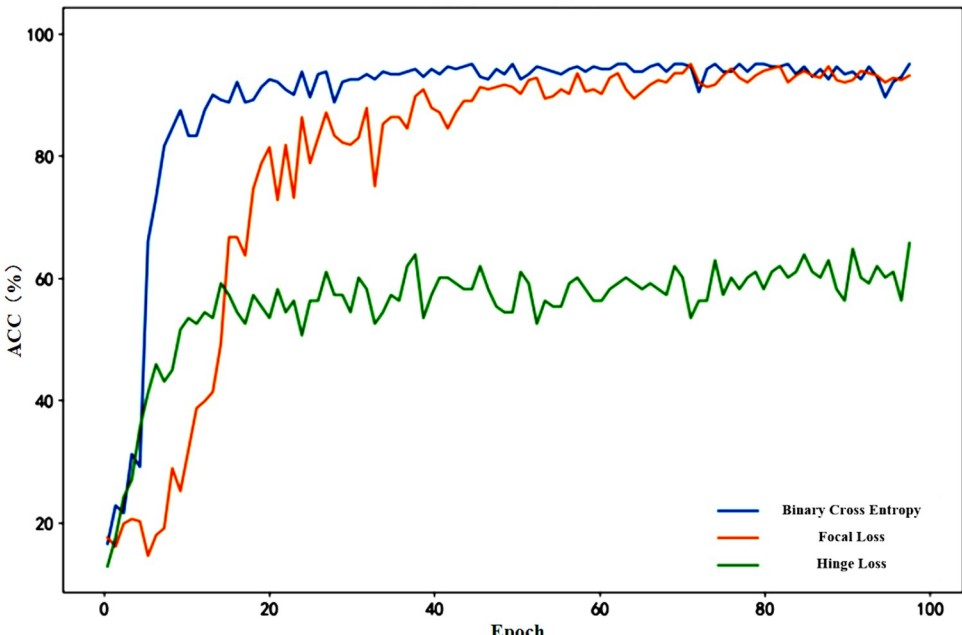

**Fig 9. The model accuracy under different loss functions.**

entropy loss has already shown a stable trend. Therefore, to save training time and improve computational efficiency, this study sets the training epochs to 50 rounds.

**Hyper-parametric analysis.** In this section, we selected three key hyper-parameters with significant impact on model performance for detailed analysis: batch size, learning rate, and the number of self-attention heads, to investigate the effects of different hyper-parameters on model performance. Different value ranges were set for each hyper-parameter, forming three experimental groups accordingly. Meanwhile, the parameters of the model proposed in this paper were set as the baseline to ensure that each experimental group controlled only one variable, enabling a more accurate observation and analysis of changes in model performance. The experimental results are shown in Table 2.

Firstly, experiments were conducted on batch size. By setting different batch size values, it was found that batch size significantly influences the model's performance. As shown in Table 2, the model's performance is optimal when the batch size is set to 32. Both larger and smaller batch sizes result in decreased model performance. This is because a too large batch size may cause the model to overly focus on the overall data distribution during training, neglecting local features. Conversely, a smaller batch size may cause the model to overly focus on local features during training, neglecting the overall data distribution.

Next, the influence of the learning rate on model performance was investigated. The learning rate determines the step size of parameter updates during training and plays a crucial role in the model's final performance. As shown in Table 2, when the learning rate is set to 0.01, although the model's sensitivity slightly decreases, its accuracy and specificity are highest, resulting in the best model performance. This is because a smaller learning rate may lead the model to get stuck in local optima.

Lastly, the influence of the number of self-attention heads on model performance was analyzed. The number of heads determines the learning capacity of multiple sub-layers and plays a crucial role in improving model performance. As shown in Table 2, when the number of

**Table 2. Performance of different hyper-parameter.**

| Parameters | Value | ACC (%) | SEN (%) | SPE (%) |
|---|---|---|---|---|
| Batch size | 16 | 83.32 | 84.26 | 83.97 |
| | **32** | **92.89** | **96.17** | **92.99** |
| | 64 | 92.14 | 94.36 | 92.28 |
| | 128 | 88.92 | 90.41 | 89.32 |
| Learning rate | **0.01** | **92.89** | **96.17** | **92.99** |
| | 0.005 | 92.37 | 96.21 | 91.53 |
| | 0.001 | 91.66 | 95.32 | 92.34 |
| Heads | 1 | 81.73 | 90.61 | 85.36 |
| | 2 | 84.06 | 94.49 | 87.86 |
| | 4 | 87.22 | 93.32 | 90.35 |
| | **8** | **92.89** | **96.17** | **92.99** |
| | 16 | 91.64 | 94.44 | 93.15 |

heads is set to 1, the model performs the worst. This is due to insufficient feature extraction caused by too few heads. As the number of heads increases, various performance metrics of the model also improve. This indicates that a moderate increase in the number of heads helps the model capture richer feature information, thereby enhancing its performance. When the number of attention heads reaches 8, both the average accuracy and sensitivity of the model reach their peak values. However, continuing to increase the number of heads to 16 results in a significant decrease in average accuracy and sensitivity, although the average specificity slightly improves. This is because an excessive number of heads doesn't always significantly improve model performance; instead, it can lead to information redundancy and mutual interference, causing a decline in model performance. Moreover, an excessive number of heads may also increase computational costs. Therefore, considering both model performance and computational costs, this paper selects 8 self-attention heads as the configuration for the model presented.

**Convolutional structure analysis of RFFN.** To further investigate the impact of combining 1x1 convolution with 3x3 depth-wise separable convolution on model performance in RFFN compared to using either convolution method alone, this paper designed two sets of control experiments. Experiment A employed a RFFN consisting solely of 1x1 convolutions. Experiment B utilized a RFFN composed solely of 3x3 depth-wise separable convolutions. The model proposed in this paper adopted a RFFN combining both convolution types. Under the same experimental conditions, the obtained results are shown in Table 3.

As shown in Table 3, the training time for Experiment A is shorter compared to Experiment B. This is mainly attributed to the use of 1×1 convolution, which is primarily used to reduce the depth of feature maps and introduce non-linearity. Its operations are relatively simple and fast because they do not involve complex spatial calculations. The 3x3 depth-wise separable

**Table 3. Performance comparison of different convolutional structure models.**

| Model | ACC(%) | SEN(%) | SPE(%) | F1(%) | AUC(%) | Training Duration(min) |
|---|---|---|---|---|---|---|
| A | 88.67 | 91.21 | 92.66 | 91.39 | 93.26 | 115.70 |
| B | 91.02 | 93.54 | 93.01 | 93.97 | 94.78 | 146.20 |
| This work | 92.89 | 96.17 | 92.99 | 94.41 | 96.77 | 127.30 |

convolution used in Experiment B requires applying a 3×3 convolution kernel on each input channel. While this increases computational complexity, it allows for more effective capture of spatial and temporal features in the signals, resulting in Experiment B out-performing Experiment A in terms of performance. Compared to models using only single convolutions, the model proposed in this paper demonstrates superior performance. This is attributed to the model's clever combination of the advantages of both convolutions, enhancing overall performance while maintaining relatively efficient computational capabilities.

## Cross-patient detection performance

Individual differences exist in epileptic EEG signals, and even within the same individual, the characterization of EEG signals can change over a significant time interval between acquisition dates. This paper assesses the model's performance across various cases using the CHB-MIT dataset. To ensure comparability, this paper conducts an experiment based on both CNN and attention mechanisms. Unlike the current method, the model's Transformer block utilizes the original, unaltered Vision Transformer (VIT) without employing any other structures. Apart from this, the experimental environments mirror those proposed in this paper. Comparative experiments were conducted on the preprocessed data obtained in this paper. The experimental results of the comparative models are shown in Table 4.

A detailed analysis of the experimental results in Table 4 reveals that although only a very small number of patients achieved performance metrics exceeding 90% simultaneously, the majority of patients had at least one metric exceeding 90%. While the average accuracy of the model didn't reach 90%, other average metrics were within the range of 90%-92%. This indicates that the model combining CNN and attention mechanisms is feasible for epilepsy detection, but the detection performance need improvement. Therefore, modifications were made based on comparative model analysis to enhance detection accuracy. The experimental results are shown in Table 5.

**Table 4. Experimental performance of comparative model.**

| Patient | ACC(%) | SEN(%) | SPE(%) | F1(%) | AUC(%) |
|---------|--------|--------|--------|-------|--------|
| Chb1 | 83.43 | 95.37 | 85.61 | 91.17 | 92.44 |
| Chb2 | 80.64 | 84.12 | 89.64 | 86.34 | 86.88 |
| Chb3 | 88.44 | 91.51 | 90.12 | 89.99 | 95.62 |
| Chb5 | 81.69 | 87.43 | 89.83 | 88.49 | 88.14 |
| Chb8 | 89.32 | 91.83 | 92.36 | 91.67 | 93.31 |
| Chb9 | 80.33 | 89.11 | 91.45 | 89.98 | 87.19 |
| Chb10 | 86.91 | 90.28 | 92.10 | 89.75 | 94.80 |
| Chb13 | 88.62 | 94.30 | 95.31 | 92.46 | 93.62 |
| Chb14 | 81.59 | 79.24 | 87.65 | 88.63 | 88.13 |
| Chb16 | 85.84 | 88.09 | 89.33 | 90.10 | 90.26 |
| Chb17 | 86.89 | 93.43 | 91.25 | 89.76 | 91.59 |
| Chb18 | 90.63 | 90.69 | 93.26 | 91.01 | 94.89 |
| Chb19 | 88.41 | 92.88 | 95.11 | 91.28 | 89.63 |
| Chb20 | 90.23 | 93.89 | 92.87 | 90.34 | 91.64 |
| Chb21 | 89.81 | 95.22 | 91.08 | 92.18 | 95.91 |
| Chb23 | 89.32 | 95.91 | 93.14 | 94.15 | 93.32 |
| AVG | 86.38 | 90.83 | 91.26 | 90.46 | 91.71 |

**Table 5. Experimental performance of this model.**

| Patient | ACC(%) | SEN(%) | SPE(%) | F1(%) | AUC(%) |
|---------|--------|--------|--------|-------|--------|
| Chb1 | 89.14 | 100.00 | 90.17 | 93.56 | 99.93 |
| Chb2 | 88.25 | 90.13 | 95.38 | 92.63 | 91.54 |
| Chb3 | 93.34 | 95.17 | 94.02 | 94.82 | 98.31 |
| Chb5 | 89.69 | 90.34 | 96.15 | 93.26 | 91.60 |
| Chb8 | 92.43 | 98.75 | 93.22 | 95.64 | 95.80 |
| Chb9 | 88.68 | 90.92 | 94.36 | 92.31 | 91.73 |
| Chb10 | 94.71 | 100.00 | 91.01 | 96.03 | 99.75 |
| Chb13 | 94.56 | 99.89 | 92.87 | 95.58 | 99.57 |
| Chb14 | 88.35 | 85.25 | 93.96 | 90.11 | 91.92 |
| Chb16 | 91.43 | 91.10 | 95.24 | 93.25 | 94.79 |
| Chb17 | 92.29 | 97.24 | 94.13 | 95.22 | 95.51 |
| Chb18 | 97.08 | 100.00 | 93.10 | 96.51 | 99.36 |
| Chb19 | 93.63 | 100.00 | 91.27 | 95.62 | 99.45 |
| Chb20 | 97.82 | 100.00 | 90.66 | 95.01 | 99.88 |
| Chb21 | 96.49 | 99.89 | 91.35 | 95.34 | 99.28 |
| Chb23 | 98.34 | 100.00 | 90.89 | 95.66 | 99.91 |
| AVG | 92.89 | 96.17 | 92.99 | 94.41 | 96.77 |

Through analysis of the experimental results in Table 5, it is evident that the proposed model in this paper exhibits high performance on datasets with significant individual differences. The average values of the model's five performance metrics all exceed 92%. Compared to the comparative models, this model shows improvements in various key evaluation metrics, especially in sensitivity, where multiple subjects achieved 100%. Additionally, several subjects also reached an AUC value of 99%. This strongly demonstrates the excellent performance of the detection model. To visually assess the stability and generalization performance of the model, a pie chart is utilized to display the various performance metrics for each patient, as illustrated in Fig 10.

From Fig 10, it can be observed that the distribution of pie chart sectors for each patient is relatively uniform across different performance metrics, with differences of only around 1%. This clearly demonstrates that the model maintains stable recognition capabilities when faced with significant individual differences in epileptic signals. This further confirms the superiority of the model's generalization performance, providing strong support for its widespread application in epileptic detection tasks. To further explore the superiority of the models, the average performance indicators of the two models are compared, as shown in Table 6.

Table 6 indicates that the model proposed in this study outperforms the comparative models in various average performance aspects. Specifically, compared to the comparative model, proposed model achieved a 6.51% improvement in average accuracy, indicating that the model can more accurately identify epileptic seizures on the same dataset. Additionally, the average sensitivity increased by 5.34%, suggesting an enhanced capability of the model in identifying epileptic seizure signals, which is crucial for timely detection and intervention of epilepsy. The average AUC also increased by 5.06%, indicating a stronger ability of the model in distinguishing between epileptic and non-epileptic signals. Moreover, the average F1 score of our model increased by 3.95%, indicating better performance in comprehensive evaluation of precision and recall, reflecting a more comprehensive performance of the model in detection tasks. The average specificity increased by 1.73%, indicating an enhanced capability of the model in

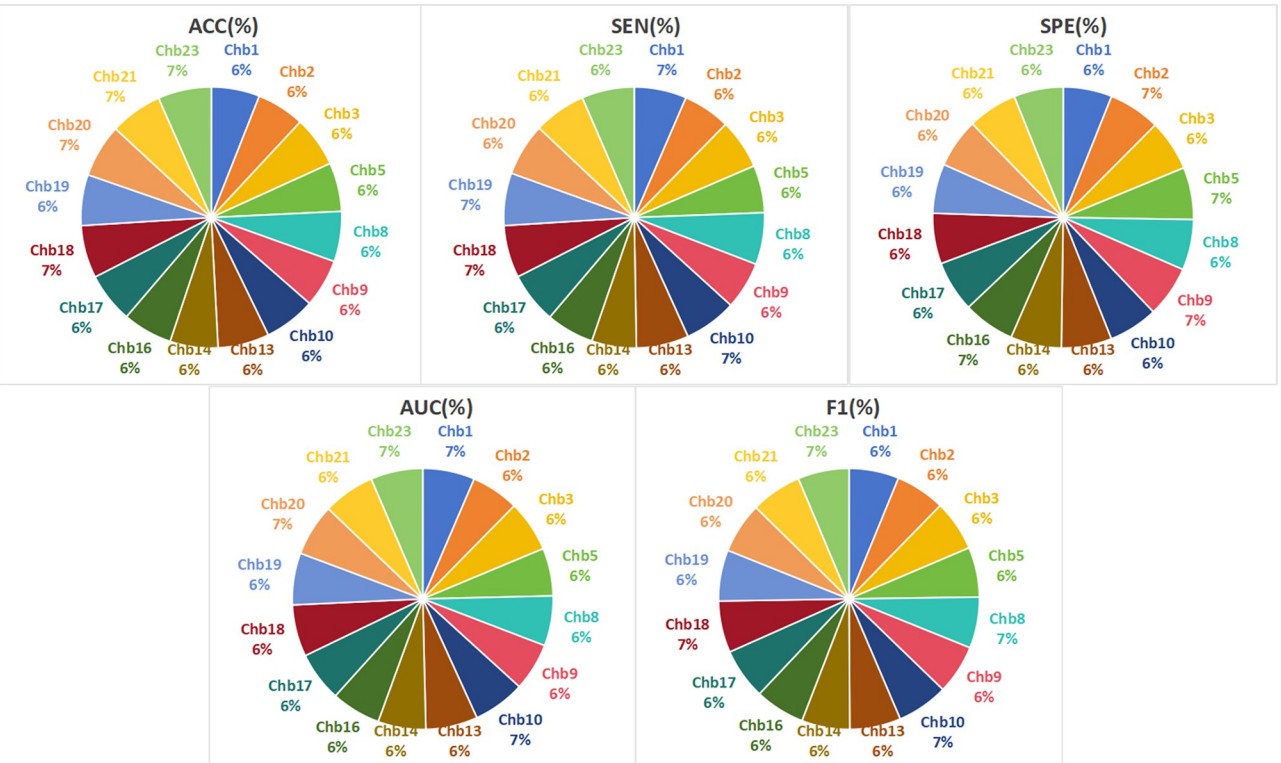

**Fig 10. Pie charts of individual patient performance indicators.**

identifying non-epileptic seizure signals, which can reduce the risk of unnecessary treatment or panic caused by false alarms for patients. Lastly, the training results demonstrate a significant reduction in training time by up to 45.01% compared to the comparative models. This data strongly proves the superiority of the "lightweight multi-head self-attention" mechanism adopted in this study in effectively reducing computational costs.

Through in-depth comparative analysis of the epileptic detection performance between proposed model and the comparative model, the following conclusions are drawn: The network structure of proposed model can more comprehensively capture key information in the data, and the optimized "light multi-head self-attention mechanism" effectively reduces computational complexity, thereby improving the overall performance of the model and reducing training time. In summary, the model successfully achieves significant savings in training time while maintaining excellent performance, fully demonstrating the perfect combination of efficiency and practicality, and providing a more efficient and practical solution for epileptic detection tasks.

**Table 6. Comparison of average performance metrics.**

| Model | ACC(%) | SEN(%) | SPE(%) | F1(%) | AUC(%) | Training Duration(min) |
|---|---|---|---|---|---|---|
| CNN+VIT | 86.38 | 90.83 | 91.26 | 90.46 | 91.71 | 231.50 |
| This work | 92.89 | 96.17 | 92.99 | 94.41 | 96.77 | 127.30 |

**Table 7. Comparison of literature.**

| Literature | Model | ACC(%) | SEN(%) | SPE(%) |
|---|---|---|---|---|
| K. Saab [27] (2020) | Dense-CNN | 72.53 | 80.53 | 68.54 |
| I. Jemal [28] (2021) | Deep-CNN | 91.82 | 91.93 | — |
| S. Yang [22] (2021) | XGBoost | 87.74 | 85.48 | 90.00 |
| Y. Zhao [29] (2022) | IBA | 76.36 | 77.42 | 76.32 |
| J. Zhou [30] (2022) | SOF | — | 84.67 | 82.06 |
| Y. Wang [31] (2023) | MAML | 71.14 | 70.07 | 72.19 |
| Z. Zhang [32] (2024) | PANN | — | 95.71 | 92.59 |
| Z. Zhang [33] (2024) | EESNN | — | 91.67 | 97.17 |
| This work | CNN+Self-Attention | 92.89 | 96.17 | 92.99 |

## Comparison of literature

To further validate the superiority of our method in the field of epilepsy detection, this paper selected a series of classical epilepsy detection methods using the CHB-MIT dataset from recent years for a detailed comparative analysis, as shown in Table 7.

As shown in Table 7, the model proposed in this study demonstrates outstanding performance in epilepsy detection tasks. In the study by Y. Wang [31], the performance of cross-patient epilepsy detection was conspicuously lacking, possibly due to training the model under limited sample conditions. The multi-view cross-target epilepsy detection model proposed by Y. Zhao et al. [29], based on Information Bottleneck Attribution (IBA), showed improved performance but still did not reach an accuracy rate of 80%. While the model by K. Saab et al. [27] achieved a sensitivity of 80.53%, its specificity was only 68.54%, showing a significant gap compared to our model. The EEG model proposed by J. Zhou et al. [30], based on Self-Organizing Fuzzy Logic (SOF) classifier, exhibited sensitivities and specificities in cross-patient detection both over 10% lower than our model. Although S. Yang et al. [22] successfully used a single model for epilepsy detection across different patients, achieving a specificity of 90%, both the accuracy and sensitivity of their model did not exceed 90%, indicating a certain gap compared to our model. In the study by I. Jemal et al. [28], they employed an improved CNN architecture based on separable deep convolution for cross-patient epilepsy detection, achieving an accuracy of 91.82% and a sensitivity of 91.93%. Compared to our model, their accuracy was 1.07% lower, and the sensitivity was 4.24% lower. This indicates that our model exhibits higher reliability in epilepsy detection tasks.

It's noteworthy that the PANN model proposed by the Z. Zhang team [32] exhibited outstanding performance in cross-patient epilepsy detection tasks, with an accuracy only 0.46% lower than our model and a specificity only 0.40% lower. Subsequently, the Z. Zhang team [33] introduced another cross-patient epilepsy detection model based on Pulse Neural Networks (EESNN). While this model showed improvement in specificity, it experienced a decline in sensitivity. Compared to our model, although the EESNN model has higher specificity, its sensitivity is lower than that of our model. When comprehensively evaluating model performance, a harmonic mean of sensitivity and specificity is typically considered. In this regard, our model demonstrates a greater advantage.

Comparatively analyzing with other classical models, our proposed model has shown relative improvements in accuracy, sensitivity, and specificity in epilepsy detection tasks.

## Conclusion

This paper proposes a cross-patient epilepsy detection method based on the multi-head self-attention mechanism. The method first preprocesses data from various patients to ensure balanced samples of different categories. Then, it performs feature fusion classification using pre-convolutional layers and multi-head self-attention modules. Additionally, the multi-head self-attention mechanism module in the model adopts an alternating structure with light multi-head attention layers, aiding in extracting more comprehensive multi-scale features while reducing computational costs. Experimental results on the CHB-MIT dataset demonstrate that the proposed model achieves accuracy, sensitivity, specificity, F1 score, and AUC of 92.89%, 96.17%, 92.99%, 94.41%, and 96.77%, respectively. The experimental results indicate that the method exhibits good stability and generalization in cross-patient epilepsy detection, which is of significant importance for assisting diagnosis and treatment.

This paper initially constructs an epilepsy detection model based on a multi-head self-attention mechanism and achieves promising results in experiments. However, there are still many aspects that require further exploration and optimization. A primary limitation in our epileptic seizure detection research stems from the reliance on commonly available public datasets for training, which, despite their widespread use, remain insufficient in terms of quantity for effective model training. Consequently, there is a need to seek out additional datasets to acquire more training data, thereby enhancing the model's generalization capabilities. In data processing, this paper employed cropping and oversampling to maintain data balance. However, these operations may actually result in information loss or redundancy, constituting the second limitation of this study. Future research needs to further optimize data processing methods, and this limitation can also be addressed with the addition of more data. Lastly, future research can further explore more efficient models by optimizing both classification performance and reducing model parameters. This can be achieved through various methods such as incorporating pre-trained models, decreasing the number of parameters in the attention modules, or pruning attention heads.

## Acknowledgments

The authors are grateful to the CHB-MIT dataset and reviewers for their valuable comments.

## Author Contributions

**Conceptualization:** Yandong Ru, Gaoyang An, Zheng Wei.

**Formal analysis:** Yandong Ru.

**Funding acquisition:** Yandong Ru.

**Methodology:** Gaoyang An.

**Project administration:** Hongming Chen.

**Software:** Yandong Ru.

**Supervision:** Hongming Chen.

**Validation:** Yandong Ru, Zheng Wei.

**Writing – original draft:** Gaoyang An.

**Writing – review & editing:** Yandong Ru, Zheng Wei, Hongming Chen.

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
