## [Decision Letter · Decision Letter 0]

17 Mar 2024

PONE-D-24-07075Epilepsy Detection Based on Multi-Head Self-Attention MechanismPLOS ONE

Dear Dr. An,

Thank you for submitting your manuscript to PLOS ONE. After careful consideration, we feel that it has merit but does not fully meet PLOS ONE’s publication criteria as it currently stands. Therefore, we invite you to submit a revised version of the manuscript that addresses the points raised during the review process.

We look forward to receiving your revised manuscript.

Kind regards,

Mustafa Sameer, Ph.D.

Academic Editor

PLOS ONE

Journal Requirements:

d) If you did not receive any funding for this study, please state: “The authors received no specific funding for this work.

"This work was supported by the Foundation for Talent of Zhejiang Ocean University (JX6311061523)."

Reviewers' comments:

Reviewer's Responses to Questions

**Comments to the Author**

1. Is the manuscript technically sound, and do the data support the conclusions?

Reviewer #1: Yes

Reviewer #2: Yes

Reviewer #3: Yes

2. Has the statistical analysis been performed appropriately and rigorously? 

Reviewer #1: Yes

Reviewer #2: No

Reviewer #3: Yes

3. Have the authors made all data underlying the findings in their manuscript fully available?

Reviewer #1: Yes

Reviewer #2: Yes

Reviewer #3: Yes

4. Is the manuscript presented in an intelligible fashion and written in standard English?

Reviewer #1: Yes

Reviewer #2: Yes

Reviewer #3: Yes

5. Review Comments to the Author

Reviewer #1: (1) The authors propose a framework that combines CNN and multi-head attention to detect epilepsy. However, combining CNN and multi-head attention is not an innovation, and much work has been done nowadays, so it is not innovative enough.

(2) The figures are not clear enough.

(3) References are not novel enough, authors should cite more literatures from the last three years.

Reviewer #2: The aim and method are concise with details but the authors could include the following comments on

1. How does the number of self-attention heads and their dimension affect the model's ability to learn intricate relationships between different feature vectors?

2. The paper introduces a "light multi-head self-attention" mechanism to reduce computational cost. How significant is this reduction compared to the original Transformer approach?

3. How does the combination of 1x1 and 3x3 depth-wise separable convolutions within the residual network impact the feature representation compared to using only one type of convolution?

4. The choice of hyper parameters for the convolutional layers and the number of self-attention heads is crucial. How sensitive is the model's performance to these choices?

5. How does the choice of loss function (e.g., cross-entropy) impact the model's training process and the final classification performance?

6. Epilepsy events are typically rare compared to normal EEG recordings. How does the model handle this class imbalance issue, and are there any specific techniques used to address it?

Reviewer #3: Please find the following comments on your paper:

1. The motivation for research is not clear from the abstract.

2. Paper contributions should be considered in the introduction section.

3. Paper organization is missing in the introduction section.

4. The related work section is missing in the paper.

5. Please improve the figure quality.

6. Please consider a system overview or block diagram of research for a better understanding of the paper.

7. Data distribution graphs should be considered for the analysis.

8. Limitations of the proposed approach should be discussed.

9. In Figs 7 and 8, the y label should be written.

6. PLOS authors have the option to publish the peer review history of their article (what does this mean?). If published, this will include your full peer review and any attached files.

Reviewer #1: No

Reviewer #2: **Yes: **Dr. M. Laavanya

Reviewer #3: **Yes: **Alwin Poulose

---

## [Author Response · Author response to Decision Letter 0]

26 Apr 2024

Revision Description

We are very grateful to the reviewers for their valuable comments, which are of great significance to our research. We have studied the reviewers’ comments and revised the paper carefully according to the comments. Changes to the original text are shown in red in the text, and new additions are shown in blue. The specific revision are as follows:

Comments from reviewer 1

1. The authors propose a framework that combines CNN and multi-head attention to detect epilepsy. However, combining CNN and multi-head attention is not an innovation, and much work has been done nowadays, so it is not innovative enough.

Revision description：

Thank you very much for your careful review and valuable feedback on our paper. Here, we would like to explain and clarify your concerns about the novelty of our work.

Firstly, this paper focuses on the research of cross-patient epilepsy detection. While CNN and self-attention mechanisms have been applied in single-patient epilepsy detection, there remains a gap in the field of cross-patient epilepsy detection. Due to the complex differences between different individuals in cross-patient detection, it is more challenging but also holds greater practical value. Therefore, this paper combines CNN with self-attention mechanisms for cross-patient epilepsy detection, contributing to further advance this field.

Additionally, this paper introduces an optimized Transformer module. The uniqueness of this module lies in its structure, which integrates CNN layers, light multi-head self-attention layers, and residual feed-forward networks, forming a new architecture distinct from traditional attention mechanism modules. This design significantly improves computational efficiency, greatly reduces training time, and enhances the overall efficiency of the model while maintaining detection performance.

2. The figures are not clear enough.

Revision description：

Due to figure format issues in the previous paper, the figures were not clear enough. This problem has now been resolved, ensuring that all figures in the paper are clear enough.

3. References are not novel enough, authors should cite more literatures from the last three years.

Revision description：

To address this issue, we have re-examined the latest research findings in the relevant field during the revision process and selected more representative and cutting-edge literature from the past three years for citation.

Please refer to the third to sixth paragraphs in the "Introduction" section and the "Comparison of literature" section in the paper for detailed changes.

Comments from reviewer 2

1. How does the number of self-attention heads and their dimension affect the model's ability to learn intricate relationships between different feature vectors?

Revision description：

Due to experimental requirements, we have integrated the section on the impact of the number of self-attention heads on the model from our previous paper into the "Hyper-parametric analysis" section of the current paper and revised its description.

Please refer to the last paragraph of the "Hyper-parametric analysis" section and Table 2 in the paper for detailed changes.

2. The paper introduces a "light multi-head self-attention" mechanism to reduce computational cost. How significant is this reduction compared to the original Transformer approach?

Revision description：

 In the comparative experiments, a new performance evaluation dimension, training time, has been added to highlight the computational efficiency of the proposed "light multi-head self-attention" mechanism. The experimental results are shown in Table 6, which comprehensively demonstrates the advantages of this mechanism in computational efficiency.

 Please refer to the last two sentences of the second-to-last paragraph in the "Cross-Patient Detection Performance" section and Table 6 in the paper for detailed changes.

 3. How does the combination of 1x1 and 3x3 depth-wise separable convolutions within the residual network impact the feature representation compared to using only one type of convolution?

Revision description：

 Added experiments named the "Convolutional structure analysis of RFFN" to the "Experimental parametric" section in this paper. The experimental results are shown in Table 5. Based on Table 5, this paper provides a detailed analysis of the impact of combining 1x1 and 3x3 depth-wise separable convolutions compared to using a single convolution type in residual networks on model performance

 Please refer to the "Convolutional structure analysis of RFFN" section in the paper for detailed changes.

 4. The choice of hyper parameters for the convolutional layers and the number of self-attention heads is crucial. How sensitive is the model's performance to these choices?

Revision description：

 Added experiments named the "Hyper-parametric analysis" to the "Experimental parametric" section in this paper. The experimental results are shown in Table 2, which can be seen that the model performance varies with different values of batch size, learning rate, and number of self-attention heads.

 Please refer to the "Hyper-parametric analysis" section in the paper for detailed changes.

 5. How does the choice of loss function (e.g., cross-entropy) impact the model's training process and the final classification performance?

Revision description：

 Added experiments named the "Loss function analysis" to the "Experimental parametric" section in this paper. The experimental results are shown in Fig 9, which can be seen that the effect of different loss functions on the performance of the model.

 Please refer to the "Loss function analysis" section in the paper for detailed changes.

 6. Epilepsy events are typically rare compared to normal EEG recordings. How does the model handle this class imbalance issue, and are there any specific techniques used to address it?

Revision description：

 To address the issue of data imbalance, we used oversampling techniques in the "Data preprocessing methods" section of our previous paper. In the current paper, we have further added a detailed description of the oversampling technique.

 Please refer to the last paragraph and the second-to-last paragraph of the "Data preprocessing methods" section in the paper for detailed changes.

Comments from reviewer 3

 1. The motivation for research is not clear from the abstract.

Revision description：

 The motivation for research in the abstract has been revised to make it clearer in this paper. 

 Please refer to the first two sentences of the "Abstract" section in the paper for detailed changes.

2. Paper contributions should be considered in the introduction section.

Revision description：

 Added the contributions of the paper to the "Introduction" section in this paper.

 Please refer to the second-to-last paragraph of the "Introduction" section in the paper for detailed changes.

 3. Paper organization is missing in the introduction section.

Revision description：

 Added the organization of the paper to the "Introduction" section in this paper.

 Please refer to the last paragraph of the "Introduction" section in the paper for detailed changes.

 4. The related work section is missing in the paper.

Revision description：

 Added the related work to the "Introduction" section in this paper.

 Please refer to the fourth to sixth paragraphs in the "Introduction" section in the paper for detailed changes.

 5. Please improve the figure quality.

Revision description：

 Due to figure format issues in the previous paper, the figures were not clear enough. This problem has now been resolved, ensuring that all figures in the paper are clear enough.

 6. Please consider a system overview or block diagram of research for a better understanding of the paper.

Revision description：

 Added the system overview of the research to the "Methods" section in this paper. And the Fig 1 in this paper shows the research block diagram.

Please refer to the first paragraph in the "Methods" section in the paper for detailed changes.

7. Data distribution graphs should be considered for the analysis.

Revision description：

 Added time-domain and frequency-domain distribution graphs of epileptic EEG signals and description to the "Data preprocessing methods" section in this paper. The Figs 6 and 7 of this paper show the distribution of epileptic EEG signals in the time and frequency domains respectively.

 Please refer to the first paragraph and the third paragraph of the "Data preprocessing methods" section in the paper for detailed changes.

 8. Limitations of the proposed approach should be discussed.

Revision description：

 Added a description of the limitations of the proposed approach to the "Conclusion" section in this paper.

 Please refer to the last paragraph of the "Conclusion" section in the paper for detailed changes.

 9. In Figs 7 and 8, the y label should be written.

Revision description：

 Due to experimental requirements, Tables 2 and 6 are currently used in this paper to present the information instead of Figs 7 and 8.

---

## [Decision Letter · Decision Letter 1]

27 May 2024

Epilepsy Detection Based on Multi-Head Self-Attention Mechanism

PONE-D-24-07075R1

Dear Dr. An,

We’re pleased to inform you that your manuscript has been judged scientifically suitable for publication and will be formally accepted for publication once it meets all outstanding technical requirements.

Kind regards,

Yuvaraj Rajamanickam, Ph.D

Academic Editor

PLOS ONE

Additional Editor Comments (optional):

Reviewers' comments:

Reviewer's Responses to Questions

**Comments to the Author**

1. If the authors have adequately addressed your comments raised in a previous round of review and you feel that this manuscript is now acceptable for publication, you may indicate that here to bypass the “Comments to the Author” section, enter your conflict of interest statement in the “Confidential to Editor” section, and submit your "Accept" recommendation.

Reviewer #1: All comments have been addressed

Reviewer #3: All comments have been addressed

2. Is the manuscript technically sound, and do the data support the conclusions?

Reviewer #1: Yes

Reviewer #3: Yes

3. Has the statistical analysis been performed appropriately and rigorously? 

Reviewer #1: Yes

Reviewer #3: Yes

4. Have the authors made all data underlying the findings in their manuscript fully available?

Reviewer #1: Yes

Reviewer #3: Yes

5. Is the manuscript presented in an intelligible fashion and written in standard English?

Reviewer #1: Yes

Reviewer #3: Yes

6. Review Comments to the Author

Reviewer #1: (No Response)

Reviewer #3: Dear Authors,

Thank you for addressing all my comments and I don't have any further concerns on the paper.

Regards

7. PLOS authors have the option to publish the peer review history of their article (what does this mean?). If published, this will include your full peer review and any attached files.

Reviewer #1: No

Reviewer #3: No

---

## [Editor Report · Acceptance letter]

31 May 2024

PONE-D-24-07075R1 

PLOS ONE

Dear Dr. An, 

I'm pleased to inform you that your manuscript has been deemed suitable for publication in PLOS ONE. Congratulations! Your manuscript is now being handed over to our production team.

Kind regards, 

on behalf of

Dr. Yuvaraj Rajamanickam 

Academic Editor

PLOS ONE